# Interdisciplinary Approach of Establishing PDAC Resectability: Biochemical, Radiological and NAT Regimen Prognostic Factors—Literature Review

**DOI:** 10.3390/medicina58060756

**Published:** 2022-06-01

**Authors:** Aiste Gulla, Daisuke Hashimoto, Doris Wagner, Ryte Damaseviciute, Kestutis Strupas, Sohei Satoi

**Affiliations:** 1Faculty of Medicine, Vilnius University, 01513 Vilnius, Lithuania; ryte.damaseviciute@mf.stud.vu.lt (R.D.); kestutis.strupas@santa.lt (K.S.); 2Department of Surgery, Institute of Clinical Medicine, Faculty of Medicine, Vilnius University, Vilnius University Hospital Santaros klinikos, 08410 Vilnius, Lithuania; 3Department of Surgery, Georgetown University Hospital, Washington, DC 20007, USA; 4Department of Surgery, Kansai Medical University, Hirakata 573-1191, Japan; daisukeh007@gmail.com (D.H.); satoi@hirakata.kmu.ac.jp (S.S.); 5Department of Visceral, General and Transplantation Surgery, University of Graz, 3, 8010 Graz, Austria; doris.wagner@medunigraz.at

**Keywords:** pancreatic adenocarcinoma, neoadjuvant treatment, surgical resectability

## Abstract

*Background and Objectives:* Pancreatic ductal adenocarcinoma (PDAC) remains one of the most lethal tumors, with a 5-year overall survival rate of less than 10%. To date, curative surgical resection remains the only favorable option for improving patients’ survival. However, there is no consensus on which prognostic biochemical, radiological markers or neoadjuvant therapy regimens would benefit patients the most. *Materials and Methods:* A literature review was performed focusing on overall survival, R0 resection, 30-day mortality, adverse events (AEs), and elevated biomarkers. The electronic databases were searched from 2015 to 2020. *Results:* We reviewed 22 independent studies. In total, 20 studies were retrospective single- or multi-center reviews, while 2 studies were prospective Phase II trials. *Conclusions:* Patients with borderline resectable or locally advanced PDAC, who received neoadjuvant therapy (NAT) and surgery, have significantly better survival rates. The CA 19-9 biomarker levels in the neoadjuvant setting should be evaluated and considered as a specific biomarker for tumor resectability and overall survival.

## 1. Introduction

Pancreatic ductal adenocarcinoma (PDAC) is one of the most lethal tumors, causing high morbidity and mortality rates each year. According to the 2018 global cancer statistics, which included data from 185 countries, the number of new PDAC cases and deaths related to the disease was nearly equal (458,918 new cases and 432,242 deaths) [1]. Despite the advances in diagnostics and treatment, the PDAC 5-year overall survival rate continues to stay under 10% [2]. An indolent disease course and non-specific symptoms account for late diagnosis at advanced stages, which disables effective treatment possibilities. Since the only curative option for PDAC is surgery, neoadjuvant therapy, especially for borderline resectable (BR) and locally advanced (LA) tumors, has become a significant part of the treatment, increasing tumor downstaging and negative margin resection. Perioperative treatment has shown positive outcomes not only in invasive tumors but also in resectable (R) cases [3,4,5]. Neoadjuvant treatment (NAT) protocols often include chemotherapy (CT) with FOLFIRINOX and Gemcitabine/nab Paclitaxel (Gem/nab-P) or chemoradiotherapy (CRT), which has been shown to improve overall survival [6,7]. Nonetheless, there is no definite agreement on the choice of drug, treatment duration, dosage, etc. that would benefit the patient the most. There are ongoing clinical trials trying to address these questions: PRO33061 (stereotactic body radiotherapy (SBRT) vs. conventionally fractionated chemoradiotherapy (CfCRT) administered preoperatively), SWOG 1505 (perioperative modified FOLFIRINOX (mFOLFIRINOX) vs. Gem/nab-P). Moreover, individuals respond differently to treatment with chemoradiotherapy or/and surgery. Thus, determining the biochemical, radiological, and NAT regimen prognostic factors is needed, which would help in terms of treatment selection in pancreatic ductal adenocarcinoma. To date, the most analyzed biomarker for diagnosis and treatment efficacy is the carbohydrate antigen 19-9 (CA19-9), which has also been shown to improve survival outcomes [8]. Several research papers have analyzed the prognostic value of pancreatic cancer treatment by combining biomarkers such as CA 19-9 with delta radiomic features, using a computed tomography scan [9]. A non-invasive approach and the ability to analyze multiple images make radiomics an appealing prognostic measure that could be implemented in the clinical practice of PDAC treatment.

This literature review was designed by reviewing single and multi-institutional studies, retrospective reviews, and systemic reviews of clinical trials to clarify valid biochemical, radiological, and NAT regimen prognostic factors for PDAC patients, which would serve as reliable prognostic factors for clinical use.

## 2. Materials and Methods

The electronic databases Medline and Scopus were searched for studies from 2015 to 2020. Twenty-two studies were included in total. Only 2 (Hayashi and Okano) of the included studies were conducted prospectively. Both were non-blinded and only one was randomized (Okano).

Subjects: male or female patients of any age with a diagnosis of PDAC and who underwent perioperative treatment (NAT) were included.

Treatment: different neoadjuvant regimen combinations were considered: chemotherapy, radiotherapy, and chemoradiotherapy followed by surgery.

Outcomes: overall survival rates, 30-day mortality rates, R0 resections, complications, and elevated biomarkers were the outcomes that were discussed in the selected articles.

Study types and design: studies were retrospective in type and prospective in design and were both multiple and single in the case of institutional reviews, studying the patients with PDAC who underwent NAT and/or followed by surgery.

Search criteria: we performed a literature search of English articles using the Medline and Scopus databases for the years 2015–2020. The search terms included PDAC, pancreatic cancer, perioperative therapy, neoadjuvant treatment, NAT, tumor size, CA 19-9, prognostic factors, pathological response, survival rate, surgical treatment, and prediction model. Title and abstract screening were performed by 2 reviewers. The included studies were saved in a Mendeley database.

Data items: the data characteristics included general information (year of publication, country, and the number of institutions); study participant characteristics (age, gender, and comorbidities (CCI); the type of neoadjuvant treatment (chemotherapy (CT) or chemoradiotherapy (CRT)); the type of surgery performed (e.g., PD or DP); inclusion and exclusion criteria, tumor characteristics (e.g., pathology reported, the size of the lesion, and T stage), main (overall survival); and secondary outcomes (30-day mortality rate, R0 resection, complications, and elevated biomarkers).

## 3. Results

Literature review flow chart (Figure 1).

### 3.1. Study Characteristics

Table 1 shows study characteristics.

In total, 22 studies were included in the literature review (Figure 1); 20 studies were retrospective single- or multi-center reviews [8,10,11,14,15,16,17,18,19,20,21,22,23,24,25,26,27,28,29,30], while 2 studies were prospective phase II trials [12,13].

#### 3.1.1. Overall Survival

There were three studies comparing the outcomes after the different neoadjuvant treatment regimens, followed by surgery. The overall survival in all the studies was significantly better (31 months vs. 13 months, 35.3 months vs. 16.3 months, 37.7 months vs. 18.6 months) in tumors treated with NAT and surgery compared with the non-resected tumor group when compared respectively; *p* = 0.03 [10,16,21]. All of the studies included FOLFIRINOX in their chemotherapy regimen. Both Barnes et al. (2019) and Michaelakos et al. (2019) used chemoradiotherapy, while Gemenetzis et al. (2019) included different forms of radiotherapy (SBRT, intensity-modulated radiotherapy/external beam radiation therapy (IMRT/EBRTB)). Including FOLFIRINOX in the perioperative chemotherapy regimens, followed by surgery, has advantages for both borderline resectable and locally advanced tumors.

Khushman et al. (2015) stated that increasing the number of full-dose FOLFIRINOX cycles for locally advanced PDAC was significantly associated with increased survival, despite the low R0 resection rate (20%) [14].

In another study by Fujii et al. (2017) neoadjuvant chemoradiotherapy (NACRT) effects were analyzed on resectable and borderline resectable tumors, stratified by portal vein involvement (BR-PV) and arterial (BR-A) involvement. Significant overall survival was only achieved in BR-PV patients, compared with patients given surgery first [15].

Whether there is a difference in overall survival and other clinical outcomes in terms of the type of radiation was analyzed by Barrord et al. (2020) [19], who compared a fractionated chemoradiation group with stereotactic body radiation therapy. The outcome was superior in the fractionated group: 47.4 months (95% CI, 24.9–61.2) versus 19.3 (95% CI, 14.7–32.2). Gemcitabine/nab-paclitaxel chemotherapy was used by the majority of patients in fractionated and SBRT groups (68% vs. 72%). FOLFIRINOX was the second most common regimen.

In the study by Aoki et al. (2020) patients were stratified into two groups (responder and non-responder) by evaluating perioperative CA 19-9 biomarker, which in the current literature is suggested as an efficient prognostic marker for overall survival and resectability [31,32]. The group who had normalized CA 19-9 had a better survival outcome [8].

Patients with borderline resectable or locally advanced PDAC, who received NAT and surgery, had significantly better survival rates compared to patients who were not resected [10,16,21]. These findings align with the study led by Rangelova et al. (2019), where resected BR and LA PDAC patients showed a significantly longer survival compared with a non-resected group (median survival of 22.4 vs. 12.7 months, respectively), irrespective of the type and dose of NAT given for both groups [3]. These findings align with the Prep-02/JSAP-05 study led by Satoi S, et al., ASCO 2019 (unpublished data), wherein the effects of neoadjuvant chemotherapy with gemcitabine and S-1 for resectable PDAC showed significantly longer survival rates with acceptable feasibility, compared to the cohort given surgery first. The most frequent causes of unresectability between studies were local disease progression and distant metastases. Interestingly, despite the different status of resectability (BR, LA, or both), all studies have shown comparable results (31 months, 35.3 months, 37.7 months). All the studies included FOLFIRINOX in their chemotherapy regimen. The treatment benefits of FOLFIRINOX have been reported in multiple studies. For instance, in the systematic review and meta-analysis of borderline resectable PDAC patients, Jansen et al. reported favorable OS results (22.2 months (95% CI% 18.8 to 25.6 months) [33], while Hackert et al. (2018) reported a > 60% tumor resectability rate in patients with borderline resectable, locally advanced and metastatic disease [34]. There are ongoing trials that are further analyzing the effect of FOLFIRINOX (NCT01560949, NCT02047474, NCT02562716, NCT01591733).

Besides various chemotherapy protocols, radiotherapy has also been used alone or in combination with chemotherapy. While the National Comprehensive Cancer Network (NCCN) version 1.2020 guidelines for PDAC consider FOLFIRINOX and gemcitabine + albumin-bound paclitaxel to be the preferred drugs, the role of radiation therapy is not so firmly defined [35]. For perioperative treatment, radiation therapy establishes a better chance for R0 resections by reducing the tumor masses around critical structures and treating regional lymph nodes. The study by Cloyd et al. analyzed two patient groups with resectable PDAC who were undergoing chemotherapy and chemoradiation. CRT significantly improved margin negative (91% vs. 79%, *p* < 0.01) and node-negative (53% vs. 23%, *p* < 0.01) resections, and overall survival was comparable with chemotherapy (OS; 33.6 vs. 26.4 months, *p* = 0.09) [36]. However, patients treated with radiochemotherapy in early-stage PDAC showed an increased 90-day mortality rate (*p* = 0.015) and shorter OS (*p* = 0.0502) than those treated with chemotherapy only [37]. In this study, chemoradiation was a common treatment of choice. Barrord et al. (2020) further analyzed the types of radiation therapy, comparing the outcomes of stereotactic body radiation therapy (SBRT) versus fractionated chemoradiation. After resection, 8 (44%) patients treated with fractionated CRT and 9 (75%) treated with SBRT developed distant (100% in the fractionated group) or local metastases. Progression-free survival and local recurrence-free survival was significantly decreased in the SBRT group vs. those receiving fractionated CRT (progression-free survival: 11 months vs. 23 months (*p* = 0.006), local recurrence-free survival at 24 months: 21% vs. 86% (*p* = 0.003). Despite the small sample size, the difference in local and distant metastases is prominent.

#### 3.1.2. Data for 30-Day Mortality, Complications, R0 Resection, Elevated Biomarkers

Barnes et al. (2019) reported a significant difference in non-resected and resected patient groups comparing elevated perioperative biomarkers. In the study by Bednar et al. (2017) significant difference of elevated biomarkers was seen in the patient group treated with the ‘‘old’’ (gemcitabine or fluorouracil (5-FU)) chemotherapy compared with the ‘‘new’’ (FOLFIRINOX and/or gemcitabine/nab-paclitaxel) [10,24].

The biomarker CA 19-9 is expressed in other cancer types, such as colorectal, gastric, and ovarian cancers. Although there might be false-positive results in benign pancreatobiliary diseases, the marker is mostly specific and sensitive to pancreatic cancer [38]. Its prognostic value was determined in neoadjuvant settings, as the CA 19-9 response was associated with R0 resection rate, histopathological response, and survival by Boone et al. [32]. Since there are multiple types of neoadjuvant regimens used in the treatment of PDAC, biochemical markers are needed to determine which patients undergoing perioperative therapy and resection could benefit the most from a particular intervention.

By measuring CA 19-9 preoperatively (during CRT and at 2 months after), Takahashi et al. classified biomarker alterations into three categories: I (increased), MD (modestly decreased), and SD (substantially decreased). PDAC patients were also divided into two efficacy groups, according to the clinical course during CRT: A (not resected after CRT because of disease progression, disease recurrence within 6 months of resection) and B (resected after CRT, disease recurrence more than 6 months after surgery). CA 19-9 alteration was an independent variable for efficacy grouping as well as for OS, which could be used in the decision-making of resectable PDAC patients undergoing neoadjuvant treatment with CRT [39]. Since the International Study Group of Pancreatic Surgery (ISGPS) and the International Association of Pancreatology (IAP) proposed adding the serum CA 19-9 level to borderline resectability as the biological factor, its evaluation before, during, and after perioperative treatment of BRPC patients should not be missed. The study by Takahashi et al. focused on determining the CA 19-9 values for biological resectability, tumor downstaging, and its importance in terms of clinical outcomes in patients treated with perioperative CRT. Pre-CA19-9 levels of 120 U/mL were set to be the minimum cutoff value that identified a subgroup of borderline resectable patients, with comparable survival results among the resectable group. Among the 58 resected BR patients who had normalized CA 19-9 (≤37 U/mL), the overall survival rate was comparable with that of patients with R disease (65%, *p* = 0.369) [40]. Barnes et al. (2019) analyzed 185 patients (115 were resected), with a focus on the change of preoperative and postoperative CA 19-9. A significant difference in overall survival was seen between the 68 patients with a normal CA 19-9 value and 47 patients with elevated biomarkers after neoadjuvant treatment (46 months vs. 25 months; log-rank *p* = 0.0003). Moreover, CA 19-9 was significantly higher in the non-resected group [10]. The reduction of CA 19-9 in the perioperative setting could serve as a prognostic factor for resectability in high-risk BR PDAC patients.

A similar approach was used in the study by Aoki et al. (2020), wherein patients were stratified into 3 groups (responder, normal, non-responder), based on their CA 19-9 response to neoadjuvant treatment with systemic chemotherapy and chemoradiotherapy [8]. A significant difference in CA 19-9 postoperatively was identified in responder and non-responder groups. Moreover, in the responder group, the CA 19-9 decrease during perioperative treatment was significantly larger and was more often normalized after tumor resection. In addition, R0 resection was more common in the responder and normal groups. As mentioned before, CA 19-9 can be elevated in other cancers and hepatobiliary diseases, thus reducing the specificity of the treatment response. Recently, Nasief et al. suggested improving treatment response prediction by using a combination of delta radiomics and the CA 19-9 antigen [9]. The combination of delta radiomics features and CA 19-9 enhanced median survival and 5-year survival probability compared with analyzing the clinical features separately (patients with increasing CA 19-9 levels: median survival (33 months) and the combination of CA 19-9 and DRF in patients with increasing CA 19-9 levels (49 months)). This approach of using several features in treatment prognosis would help to reduce false-positive CA 19-9 results, thus enabling better decisions in terms of treatment and improving patient outcomes. There are several trials supporting neoadjuvant therapy as the treatment of choice for resectable and borderline resectable PDAC patients providing favorable outcomes.

A randomized phase II/III trial (Prep-02/JSAP05) by Motoi et al. suggests that neoadjuvant therapy with gemcitabine and S-1 (GS) is a superior treatment for potentially resectable PDAC. The primary endpoint for phase III was overall survival (OS), which was significantly higher in the NAC-GS arm versus Up-S (36.7 months vs. 26.6 months; HR 0.72) (95% confidence interval 0.55–0.94; log-rank *p* = 0.015). The R0 resection rates and morbidity of the operation were comparable in both arms of the study [40]. Another phase-III trial (PREOPANC) by Versteijne et al. stratified resectable and borderline-resectable PDAC patients in the two arms of the study: one undergoing perioperative chemoradiotherapy with gemcitabine and the other an upfront resection. Although OS was not significant (16.0 months with preoperative chemoradiotherapy and 14.3 months with immediate surgery (hazard ratio, 0.78; 95% CI, 0.58 to 1.05; *p* = 0.096), the R0 resection rate was 71% in the chemoradiotherapy arm compared with 40% of immediate surgery patients (*p* < 0.001) [41].

A single-arm phase-II study carried out by Takahashi et al. analyzed the effect of neoadjuvant S-1 and concurrent radiotherapy followed by surgery in borderline resectable pancreatic cancer. CRT was completed in 96% of patients. The 2-year OS rate and median OS rate of 41 BRPC patients were 58% and 30.8 months respectively, demonstrating the efficacy of the neoadjuvant treatment for borderline resectable patients [42].

#### 3.1.3. Tumor Angiogenesis, Immunosuppression, and the Microenvironment

Since Folkman et al. proposed the hypothesis that tumor growth depends on angiogenesis, angiogenic regimens have been implemented in the treatment of multiple solid tumors. PDAC is known to be a hypoxic/fibrotic tumor, which presents rich stroma and secretes proangiogenic factors, namely, VEGF, MMP-9, interleukin-8, and fibroblast growth factor-2 [43]. Moreover, pancreatic cancer is considered to have an exceedingly immunosuppressive environment. One of the substances found in the extracellular matrix is hyaluronan, which promotes cell proliferation, migration, invasion, metastasis, angiogenesis, and chemotherapeutic resistance. Thus, it could be a novel target in cancer therapy. Interaction between epithelial, tumor, and stromal cells is the key factor in the tumor microenvironment. TME substances can induce angiogenesis and the development of cancer and metastasis, whereas the immune cells found in TME, namely, CTLA-4, PD1, PD-L1, LAG3, and VISTA participate in the antitumor immune response. This is an example of the close interconnection between angiogenesis and tumor suppression factors existing in TME and the complexity of the PDAC pathophysiology. The focus should be continued on evolving targeted therapy to reach favorable outcomes in PDAC management. The tumor microenvironment, especially in pancreatic ductal adenocarcinoma, can have a significant impact on the success of targeted therapy. Immune cells, pancreatic stellate cells, acellular stroma, and soluble factors hinder immune responses. The focus on targeted therapy shows promising results in anti-cancer treatment. For example, the inhibition of the FGFR signaling pathway was shown to be a successful cancer treatment option, since the overexpression of FGFR receptors contributes to angiogenesis and mitogenesis. Meanwhile, VEGFR overexpression is responsible for notable microvessel density; thus, the targeted inhibition of VEGFR can hinder the metastasis and growth of PDAC [44]. Another example is Galectin-1, which, when produced in large quantities, causes an immunosuppressive microenvironment and an increase in angiogenesis [45,46]. Thus, targeting this factor could improve the T-cell response and reduce tumor size. One of the ways to overcome resistance to antiangiogenic therapies is administering multiple anti-angiogenic molecules or a combination of anti-angiogenic drugs with other treatment regimens. The immunosuppressive PD-1/PDL-1 signaling mechanism is actively investigated in anti-angiogenesis-resistant tumors. Cathepsins and galectins also act as pro-angiogenic factors, thus creating a niche for targeted immunotherapy to improve the treatment response.

## 4. Conclusions

This review included articles analyzing biochemical, radiological, and NAT regimen prognostic factors for establishing PDAC resectability. Patients receiving neoadjuvant therapy with chemoradiotherapy following resection show significant overall survival and resectability rates; thus, the level of CA 19-9 should be considered during preoperative planning.

## Figures and Tables

**Figure 1 medicina-58-00756-f001:**
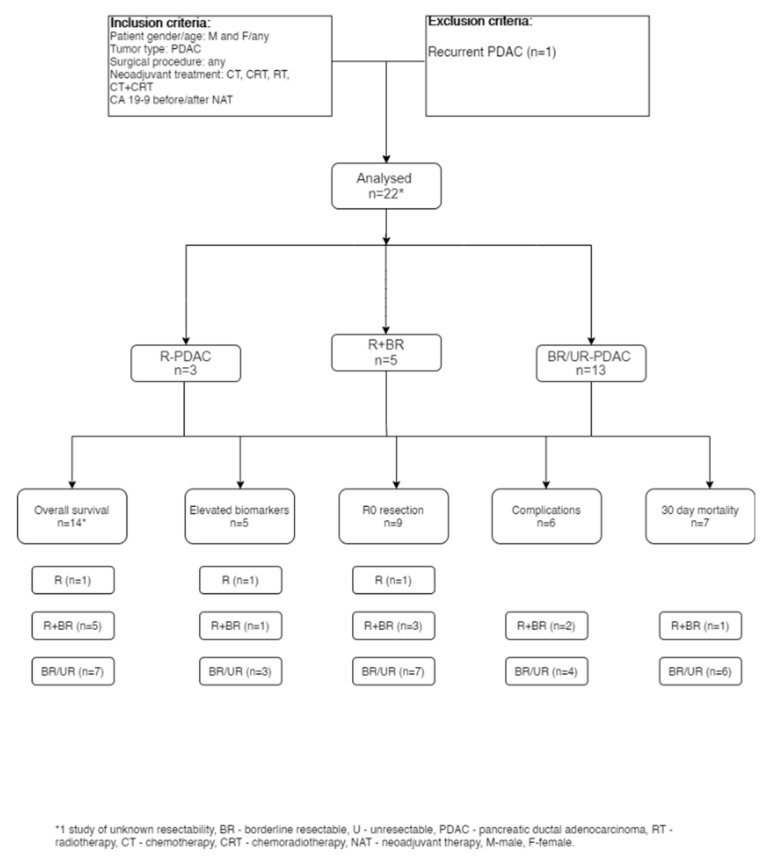
Literature review flow chart.

**Table 1 medicina-58-00756-t001:** Study characteristics of 22 studies included in the literature review.

Study (Refference)	Country	Evidence	NAT Regimen	Resectability	Multi- vs. Single-Center
Barnes et al., 2019 [10]	US	Retrospective	(A) Chemotherapy with 5-fluorouracil, leucovorin, irinotecan and oxaliplatin (FOLFIRINOX), or gemcitabine with nab-paclitaxel/2 months Chemoradiation with gemcitabine (300–400 mg/m^2^ at fixed dose rate, infused weekly x 6) or capecitabine (825 mg/m^2^ orally twice daily during the radiation therapy, RT dose 50.4Gy)/5.5 weeks (C) Both treatments	BR	Single
Xiang M et al., 2020 [11]	US	Retrospective	(A) CT; *n* = 1355 (B) CT and CFRT; *n* = 552 RT dose 50.4 Gy in 28 fractions (C) CT and SBRT; *n* = 175 RT dose 35 Gy in 5 fractions	R	Multi
Hayashi T. et al., 2019 [12]	Japan	Prospective phase II	Upfront S1-based CRT ((total dose of 50.4 Gy in 28 fractions and orally administered S-1 at 80 mg/m^2^ on the day of irradiation)/6 weeks; *n* = 45) + GEM (administration of gemcitabine at 1000 mg/m^2^ dose on days 1, 8, and 15 in 3 cycles)/12 weeks; *n* = 36	BR	Multi
Okano K. et al., 2017 [13]	Japan	Prospective phase II	NACRT + S1/hypofractionated external-beam radiotherapy (30 Gy in 10 fractions) with concurrent S-1 5 days/week for 2 weeks	R, BR	Single
Khushman et al., 2015 [14]	US	Retrospective	CT full dose FOLFRINOX 8 cycles (2–29) (*n* = 51) + CCRT (*n* = 26)	LA	Multi
Fujii et al., 2017 [15]	Japan	Retrospective	(A) NACRT-Resectable PDAC; *n* = 40 (B) NACRT-BR-PV PDAC; *n* = 27 (C) NACRT-BR-A PDAC; *n* = 21 Radiation therapy (50.4 gray in 28 fractions) combined with systemic chemotherapy involving oral S-1, the oral 5-fluorouracil pro-drug tegafur, with oteracil and gimeracil. S-1 was orally administered twice daily (80 mg/m^2^/day) from days 1 to 14 and from days 22 to 35.	R, BR	Multi
Gemenetzis et al., 2019 [16]	US	Retrospective	(A) FFX-based (B) FFX-gemcitabine combination (C) Gemcitabine-based (D) SBRT (E) IMRT/EBRTB	LA	Single
Yamada et al., 2020 [17]	Japan	Retrospective	(A) S-1 (oral 5-fluorouracil prodrug tegafur + oteracil and gimeracil) + radiotherapy and gemcitabine + S-1 (B) Gemcitabine + nab-paclitaxel and FOLFIRINOX/DP 24 (18) + radiation therapy; *n* = 61 (45%) 2.7 mo (0–23)	R, BR, Unresectable (U)	
Wolfe et al., 2020 [18]	US	Retrospective	(A) modified FOLFIRINOX; *n* = 119 (oxaliplatin, 85 mg/m^2^, irinotecan, 165–180 mg/m^2^ leucovorin, followed by 2400 mg/m^2^ 46-hour continuous infusion, all given every 2 weeks for 4–8 cycles total/3.0 cycles (2, 8)) (B) nab-P/G; *n =* 56 (biweekly regimen of 1000 mg/m^2^ gemcitabine combined with nab-paclitaxel, 125 mg/m^2^, administered on days 1 and 15 of a 28-day cycle)/3.0 cycles (2, 6) Neoadjuvant RT (A) 34 (65.4%) (B) 11 (55.0%)	BR, LA	Single
Barrord et al., 2020 [19]	US	Retrospective	(A) Fractionated Chemoradiation; *n* = 25 + CT FOLFIRINOX 7 (28) Gemcitabine/nab-paclitaxel 17 (68) Gemcitabine/cisplatin 1 (4) (B) SBRT *n* = 18 + CT FOLFIRINOX 5 (28) Gemcitabine/nab-paclitaxel 13 (72) Gemcitabine/cisplatin 0	R, BR	Single
Panni et al., 2018 [20]	US	Retrospective	7.6% of patients received neoadjuvant FOLFIRINOX 26.6% of patients received Gemcitabine plus Abraxane 20.9% of patients received Gemcitabine alone 2.9% of received other 5-FU based regimens Neoadjuvant radiation therapy in 96.2% of patients	-	Single
Aoki et al., 2019 [8]	Japan	Retrospective	Systemic chemotherapy; *n* = 27 (55%) Chemoradiotherapy; *n* = 22 (45%)	R, BR	Multi
Michelakos et al., 2019 [21]	US	Retrospective	FOLFIRINOX (with 5-FU administered as a bolus of 400 mg/m^2^, a bolus of leucovorin 400 mg/m^2^, followed by continuous infusion at 1200 mg/m^2^ per day for 46 h, oxaliplatin 85 mg/m^2^, and irinotecan 180 mg/m^2^ Chemoradiation following FOLFIRINOX, when administered, consisted of 5-FU or capecitabine with 50.4 Gy photon or 25 Gy proton radiotherapy	BR, LA	Single
Dhir et al., 2018 [22]	US	Retrospective	(A) FOLFRINOX/3 cycles (3–4); *n* = 73 (B) G-nP/2 cycles (2–3); *n* = 120	R, BR	Single
Macedo et al., 2019 [23]	US	Retrospective	(A) FOLFIRINOX/5 cycles (4–6) 66.7% + RT (SBRT 6.3% or conventional 19.3%) (B) GNP/3 cycles (2–4) 33.3% + RT (SBRT 6.6% or conventional 29.7%)	BR, LA	Multi
Bednar et al., 2017 [24]	US	Retrospective	(A) FOLFIRINOX (34%), gemcitabine/nab-paclitaxel (33%)/FOLFIRINOX and gemcitabine/nab-paclitaxel (33%)/2 lines of therapy (1–5); *n* = 61 (B) Gemcitabine or 5-FU/1 line of therapy (1–3); *n* = 31 (100%)	LA	Single
Abbas et al., 2020 [25]	US	Retrospective	Gem (vs 5-FU) + RT (SBRT), Other	R, BR, LA	Single
Sell et al., 2020 [26]	US	Retrospective	Neoadjuvant Chemotherapy Neoadjuvant radiation	R, BR, LA	Single
Kizy et al., 2020 [27]	US	Retrospective	(A) Gemcitabine; *n* = 17 (47%) (B) Multi-agent (Gem/nab-paclitaxel, FOLFIRINOX); *n* = 19 (53%) /88 days	R	Single
Yoo et al., 2017 [28]	South Korea	Retrospective	(A) FOLFIRINOX 6 cycles (3–13) (B) FDR-GEM plus CAP (3–6) Intravenous gemcitabine at 1250 mg/m^2^ with a 10 mg/m^2^/min infusion rate on days 1 and 8, and oral capecitabine at 950 mg/m^2^ twice daily on days 1–14, every 3 weeks	BR	Single
Mokdad et al., 2018 [29]	US	Retrospective	(A) CRT; *n* = 710 (B) CT; *n* = 616	R	Multi
Blair et al., 2018 [30]	US	Retrospective	(A) CT (n) (Gemcitabine alone (2), FOLFIRINOX (44), other multiagent (15) + SBRT (33 Gy in 5 fractions)/6–8 weeks; *n* = 61 (36%) (B) CT (n) (Capecitabine alone (18), 5-fluorouracil alone (2), Gemcitabine alone (7), FOLFIRINOX (26), other multiagent (54) + CRT (45 to 54 Gy in 28 fractions) GNP or FOLFRINOX; *n* = 107 (64%)	BR, LA	Single

## Data Availability

The data presented in this study are available on request from the corresponding author. All literature search data used in the manuscript are available at the Institute of Clinical Medicine, Faculty of Medicine, Vilnius, Lithuania.

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
