# Peer review of "Interdisciplinary Approach of Establishing PDAC Resectability: Biochemical, Radiological and NAT Regimen Prognostic Factors—Literature Review"

_medicina, 2022, doi:10.3390/medicina58060756_

Round 1
Reviewer 1 Report
The topic of review is very actual for oncologists and surgeons dealing with PDAC patients when curative intent surgery is considered. The implementation of biological (biochemical) factors, not only structural, seems to be more important for avoiding of nonbeneficials resections. The clinical practice has no other biochemical marker than CA 19-9 so far. The review gives the comprehensive information about results of retrospective and 2 prospective studies with neoadjuvant therapy, mostly in BR and LA PDAC. The manuscript is written in accordance with MDPI guidelines for reviews. All parts - abstract, introduction, methods results and conclusion are logically defined.
In the References chapter ref no 8 should be checked and completed.
Author Response
The topic of review is very actual for oncologists and surgeons dealing with PDAC patients when curative intent surgery is considered. The implementation of biological (biochemical) factors, not only structural, seems to be more important for avoiding of nonbeneficials resections. The clinical practice has no other biochemical marker than CA 19-9 so far. The review gives the comprehensive information about results of retrospective and 2 prospective studies with neoadjuvant therapy, mostly in BR and LA PDAC. The manuscript is written in accordance with MDPI guidelines for reviews. All parts - abstract, introduction, methods results and conclusion are logically defined.
In the References chapter ref no 8 should be checked and completed.
Response: Thank you for your remark. The reference no 8 was checked and corrected according to ACS style. (line 273-278)
Reviewer 2 Report
The authors reviewed an interdisciplinary approach of astablishing PDAC resectability.
Points to be clarified:
1) The rationale of why the authors came up with this review.
2) What is the information that is not exactly available that motivated the authors to come up with this information. What are the current caveats and how do the authors highlight the current research in answering them? If not they need to address it in future directions.
3)As is now well known, tumors grow and evolve through a constant crosstalk with the surrounding microenvironment, and emerging evidence indicates that angiogenesis and immunosuppression frequently occur simultaneously in response to this crosstalk: please expand.
4) In the frame of point 3 thinking, strategies combining anti-angiogenic therapy and immunotherapy seem to have the potential to tip the balance of the tumor microenvironment and improve treatment response (please refer to PMID: 33918146 and expand).
5. Moreover, e association of multiple anti-angiogenic molecules or a combination of anti-angiogenic drugs with other treatment regimens have been indicated as alternative therapeutic strategies to overcome resistance to anti-angiogenic therapies. Alternative mechanisms of tumor vasculature, including intussusceptive microvascular growth (IMG), vasculogenic mimicry, and vascular co-option, are involved in resistance to anti-angiogenic therapies. The crosstalk between angiogenesis and immune cells explains the efficacy of combining anti-angiogenic drugs with immune check-point inhibitors. Collectively, in order to increase clinical benefits and overcome resistance to anti-angiogenesis therapies, pan-omics profiling is key (please refer PMID: 34298648).
6. A workflow scheme or a graphical abstract might improve the manuscript quality.
Author Response
Reviewer 2
1) The rationale of why the authors came up with this review.
Response: Thank you for your remark. Our rationale for this review was to give a comprehensive information about effects of neoadjuvant therapy, mostly in BR and LA pancreatic ductal adenocarcinoma and to discuss currently available and evolving biochemical, radiological and NAT regimen prognostic factors. As of our knowledge, most reviews are limited to one type of treatment. In our work we have tried to review multiple aspects of PDAC treatment planning and efficacy.
2) What is the information that is not exactly available that motivated the authors to come up with this information. What are the current caveats and how do the authors highlight the current research in answering them? If not they need to address it in future directions.
Response: Thank you for your remark. We believe that there is a lack of systemic evaluation of PDAC resectability, involving available multiple biochemical and radiological markers, that would alleviate the decision making in PDAC management. Future goals is to do a systematic review and meta-analysis, which could clearly distinguish the most reliable markers in PDAC resectability.
3)As is now well known, tumors grow and evolve through a constant crosstalk with the surrounding microenvironment, and emerging evidence indicates that angiogenesis and immunosuppression frequently occur simultaneously in response to this crosstalk: please expand.
Response: Thank you for your suggestion. Since Folkman et al. proposed the hypothesis that tumor growth depends on angiogenesis, angiogenic regimens have been implemented in the treatment of multiple solid tumors. PDAC is known to be a hypoxic/fibrotic tumor, which has rich stroma and secretes proangiogenic factors namely VEGF, MMP-9, interleukin-8, and fibroblast growth factor-2. (1) Moreover, pancreatic cancer is considered to have an exceedingly immunosuppressive environment. One of the substances found in extracellular matrix is hyaluronan, which promotes cell proliferation, migration, invasion, metastasis, angiogenesis and chemotherapeutic resistance. Thus, it could be a novel target in cancer therapy. Interaction between epithelial, tumor and stromal cells is the key point in tumor microenvironment. TME substances can induce angiogenesis and the development of cancer and metastasis whereas immune cells found in TME namely CTLA-4, PD1, PD-L1, LAG3, VISTA participates in antitumor immune response. This is an example of close interconnection between angiogenesis and tumor suppression factors existing in TME and the complexity of the PDAC pathophysiology. A focus should be continued to evolve targeted therapy to reach favorable outcomes in PDAC management.
4) In the frame of point 3 thinking, strategies combining anti-angiogenic therapy and immunotherapy seem to have the potential to tip the balance of the tumor microenvironment and improve treatment response (please refer to PMID: 33918146 and expand).
Response: Thank you for your suggestion. Tumor microenvironment especially in pancreatic ductal adenocarcinoma can have a significant impact on the success of targeted therapy. Immune cells, pancreatic stellate cells, acellular stroma, and soluble factors hinder immune responses. The focus on targeted therapy shows promising results in anti-cancer treatment. For example, inhibition of FGFR signaling pathway was shown to be a successful cancer treatment option, since overexpression of FGFR receptors contribute to angiogenesis and mitogenesis. Meanwhile, VEGFR overexpression is responsible for great microvessel density, thus targeted inhibition of VEGFR can hinder the metastasis and growth of PDAC. (2) Another example is Galectin-1, which produced in big quantities causes and immunosuppressive microenvironment and increase in angiogenesis. (3) Thus, targeting this factor could improve T-cell response and reduce tumor size.
- Moreover, association of multiple anti-angiogenic molecules or a combination of anti-angiogenic drugs with other treatment regimens have been indicated as alternative therapeutic strategies to overcome resistance to anti-angiogenic therapies. Alternative mechanisms of tumor vasculature, including intussusceptive microvascular growth (IMG), vasculogenic mimicry, and vascular co-option, are involved in resistance to anti-angiogenic therapies. The crosstalk between angiogenesis and immune cells explains the efficacy of combining anti-angiogenic drugs with immune check-point inhibitors. Collectively, in order to increase clinical benefits and overcome resistance to anti-angiogenesis therapies, pan-omics profiling is key (please refer PMID: 34298648).
Response: Thank you for your remark. One of the ways to overcome resistance to antiangiogenic therapies is administering multiple anti-angiogenic molecules or a combination of anti-angiogenic drugs with other treatment regimens. Immunosuppressive PD-1/PDL-1 signaling mechanism is actively investigated in anti-angiogenesis resistant tumors. Cathepsins and galectins also act as pro-angiogenic factors, thus creating a niche for targeted immunotherapy to improve the treatment response. (3)
- A workflow scheme or a graphical abstract might improve the manuscript quality.
Response: Thank you for your suggestion. A workflow scheme of selected papers is provided in the manuscript page 3 and main findings of the articles in table 1 of the manuscript.
- Zhang, Z.; Ji, S.; Zhang, B.; Liu, J.; Qin, Y.; Xu, J.; Yu, X. Role of Angiogenesis in Pancreatic Cancer Biology and Therapy. Biomedicine & Pharmacotherapy, 2018, 108, 1135–1140. https://doi.org/10.1016/j.biopha.2018.09.136.
- Javadrashid, D.; Baghbanzadeh, A.; Derakhshani, A.; Leone, P.; Silvestris, N.; Racanelli, V.; Solimando, A.G.; Baradaran, B. Pancreatic Cancer Signaling Pathways, Genetic Alterations, and Tumor Microenvironment: The Barriers Affecting the Method of Treatment. Biomedicines 2021, 9, 373. https://doi.org/10.3390/biomedicines9040373
- Ribatti, D.; Solimando, A.G.; Pezzella, F. The Anti-VEGF(R) Drug Discovery Legacy: Improving Attrition Rates by Breaking the Vicious Cycle of Angiogenesis in Cancer. Cancers 2021, 13, 3433. https://doi.org/10.3390/cancers13143433
Round 2
Reviewer 2 Report
Dear Office,
The authors have clarified several of the questions I raised in my previous review. The major problems have been addressed by this revision. The manuscript can be accepted from my point of view. Best